# TablEye: Seeing small Tables through the Lens of Images

## Abstract

The exploration of few-shot tabular learning becomes imperative. Tabular data is a versatile representation that captures diverse information, yet it is not exempt from limitations, property of data and model size. Labeling extensive tabular data can be challenging, and it may not be feasible to capture every important feature. *Few-shot tabular learning*, however, remains relatively unexplored, primarily due to scarcity of shared information among independent datasets and the inherent ambiguity in defining boundaries within tabular data. To the best of our knowledge, no meaningful and unrestricted few-shot tabular learning techniques have been developed without imposing constraints on the dataset. In this paper, we propose an innovative framework called **TablEye**, which aims to overcome the limit of forming prior knowledge for tabular data by adopting domain transformation. It facilitates domain transformation by generating tabular images, which effectively conserve the intrinsic semantics of the original tabular data. This approach harnesses rigorously tested few-shot learning algorithms and embedding functions to acquire and apply prior knowledge. Leveraging shared data domains allows us to utilize this prior knowledge, originally learned from the image domain. Specifically, TablEye demonstrated a superior performance by outstripping the TabLLM in a *4*-shot task with a maximum *0.11* AUC and a STUNT in a *1*-shot setting, where it led on average by *3.17%* accuracy.

## 1 Introduction

It is a common misperception that a large volume of data is indispensable for the deep learning techniques (Zhang et al., 2018). Indeed, dataset size plays a critical role in enhancing model performance(Sordo & Zeng, 2005; Prusa et al., 2015). Regardless of a neural network model quality, it seems futile without access to ample data. This data insufficient problem frequently arises due to some reasons such as high costs, privacy concerns, or security issues(Clements et al., 2020). Despite these challenges, there are many attempts to improve accuracy through deep learning with limited labeled data. This line of research is known as few-shot learning (Wang et al., 2020).

Few-shot learning in the tabular domain, however, has received relatively little attention(Guo et al., 2017; Zhang et al., 2019). The lack of research in this area can be traced back to several factors. Firstly, compared to the image and language domain, tabular datasets lack shared information (Mathov et al., 2020). Unlike image or language data, where prior knowledge can be learned from related examples within the different datasets(Parnami & Lee, 2022), it is challenging to establish similar relationships in tabular data. For example, while learning to distinguish between dogs and lions may assist in distinguishing between cats and tigers, learning to predict solar power generation will not necessarily aid in understanding trends in the financial market. Secondly, defining clear boundaries for tabular data is a complex task(Mathov et al., 2020). Image and language data possess physical or visual representations, allowing their boundaries to be defined by parameters such as pixels, color channels (R, G, B), image size, vocabulary(words), and grammar. In contrast, tabular data lacks a distinct shared representation(Ucar et al., 2021). Various features within tabular data have independent distributions and ranges, and missing values may be present.

We assume when the features in tabular data are condensed into a limited format like pixels or words, prior knowledge learned in the different domain can help solve any task in tabular domain. For an intuitive example, if a child learns about an apple from a picture, they can connect it to a letter

('Apple') and a number ('An' or 'One') and make associations. If additional information, such as rules or relationships between numbers, is provided, the child can infer that two apples are present by observing two apple photos side by side. However, for a child who has only learned the numbers '1' and '2', understanding that 1 + 1 equals 2 may not come easily. Similarly, if we incorporate information about tabular data into neural networks trained solely on images, even a small labeled data can yield superior performance compared to traditional machine learning approaches that rely on larger amounts of labeled data.

To empirically validate our proposed hypothesis, we present the TablEye framework, which is fundamentally structured into two distinct stages. The first is the transformation stage, where each vector from a tabular dataset is transmuted into an image format. In this stage, we leverage spatial relations across three channels to ensure the tabular data not only morphs into a format analogous to conventional images but also retains its intrinsic meaning. The second stage is dedicated to the incorporation of prior knowledge through a few-shot learning approach. Recognizing the proven efficacy of few-shot learning algorithms in the realm of image processing, we capitalize on them after transforming the tabular data into an image-like structure. This transformation facilitates the construction of prior knowledge using a vast array of image data for few-shot tabular learning. Consequently, utilizing this accumulated prior knowledge enables us to predict outcomes from the image-represented tabular data effectively.

Our proposed approach achieves comparable or even superior performance to previous research through experiments on various datasets. Moreover, it offers the flexibility to perform few-shot learning tasks without being constrained by composition of dataset. TablEye overcomes the need for large unlabeled datasets by leveraging the image domain, and it requires less computing cost due to its smaller model size than one of the LLM. To the best of our knowledge, this paper represents the first attempt to apply prior knowledge from the image domain to few-shot learning in the tabular domain. The proposed few-shot tabular learning technique has the potential to provide artificial intelligence models that can achieve accurate results with only a small amount of data in scenarios where data acquisition is challenging, such as disease diagnosis in the medical industry.

The main contributions of this work are:

- This work represents the first attempt to leverage large image data as prior knowledge to address the problem of few-shot tabular learning, formation of prior knowledge.

- We propose a novel framework, TablEye, which employs domain transformation to apply prior knowledge from image data to few-shot tabular learning.

- We have successfully overcome the limitations associated with existing few-shot tabular learning models, including constraints related to feature size of dataset, the requirement for large quantities of unlabeled data, and the demand for extensive computational resources.

## 2 RELATED WORK

Tabular learning refers to the process of learning the mapping between input and output data using tabular data(Borisov et al., 2022). Tabular data is often also called structured data(Ryan, 2020) and is a subset of heterogeneous data presented in a table format with rows and columns. Each feature in this data is composed of either categorical or numerical features. Currently, methods based on decision trees and those based on Multi-Layer Perceptrons (MLP) are showing almost equal performance. Tabular learning still requires a large amount of labeled data. In the image domain, few-shot learning can easily acquire prior knowledge using many related images. For example, ProtoNet (Prototypical Network)(Snell et al., 2017) learns using similarities between images, and MAML (Model-Agnostic Meta-Learning)(Finn et al., 2017) quickly adjusts the model across various tasks, enabling rapid learning with limited data. However, in the tabular domain, there are no equivalent sets of related tabular data. Therefore, few-shot tabular learning faces significant challenges in forming prior knowledge. Therefore, the current state-of-the-art (SOTA) methods for few-shot tabular learning utilize semi-few-shot learning approaches using unlabeled data samples or transfer tabular data to the text domain and employ Large Language Models.

## 2.1 SEMI-FEW-SHOT TABULAR LEARNING: STUNT

STUNT(Nam et al., 2023) represents a semi-few-shot learning technique aimed at enhancing the performance of tabular learning in scenarios with sparse labeled datasets, utilizing a substantial quantity of reasonably coherent unlabeled data. This method marks an attempt to resolve the few-shot learning problem from a data perspective, by learning prior knowledge from an unlabeled set to which arbitrary labels have been assigned. To generate these arbitrary labels, it adopted the $k$-means clustering technique. This approach utilizes a Prototypical Network(Snell et al., 2017) to learn prior knowledge from these self-generated tasks, and it has demonstrated impressive performance. This method, as a semi-few-shot learning technique, operates exclusively within the tabular domain, by the way requires a substantial quantity of reasonably consistent unlabeled data. The size of the unlabeled set also can significantly influence the performance of STUNT(Nam et al., 2023).

## 2.2 FEW-SHOT TABULAR LEARNING: TABLLM

In the domain of few-shot tabular learning, TabLLM(Hegselmann et al., 2023) offers a unique perspective by harnessing the Large Language Model (LLM). The process employed by this method involves the conversion of original tabular data into a text format following a specific template.This transformation reformats tabular data into a more adaptable textual form, making it suitable as the prompt for LLM. Following the serialization, this data is utilized to fine-tune the LLM(Liu et al., 2022). The T0 encoder-decoder model, equipped with an extensive set of 11 billion parameters, plays a crucial role in this process (Sanh et al., 2021). This large parameter set, indicative of the extensive model training, also necessitates substantial computational resources, presenting a potential challenge. Moreover, TabLLM inevitably requires significant feature names, and t is constrained by limitations on token length.

## 3 OUR APPROACH: TABLEYE

### 3.1 OVERVIEW

This paper introduces a novel framework called TablEye, aimed at enhancing the effectiveness of few-shot tabular learning. Figure 1 shows the overview of TablEye. TablEye applies efficient few-shot learning algorithms in the image domain by performing domain transformation from the tabular domain to the image domain. The framework comprises two main stages: the transformation stage from the tabular domain to the image domain and the prior knowledge learning stage in the image domain. In the tabular domain, TablEye preprocesses tabular data and undergoes a transformation process into a three-channel image, referred to as a **tabular image**. Subsequently, few-shot tabular classification is performed using prior knowledge learned from mini-ImageNet in the image domain. To generate tabular images from tabular data, a method based on feature similarity is employed, incorporating spatial relations into the tabular images. In the stage of learning prior knowledge, ProtoNet (Prototypical Network) and MAML (Model-Agnostic Meta Learning) are employed, as they demonstrate high performance and can be applied to various few-shot learning structures. The backbone for embedding and the classifier for the few-shot task are connected sequentially. During the process of learning embeddings in a dimension suitable for classification through the backbone, Cross-entropy loss is utilized(Zhang & Sabuncu, 2018).

### 3.2 DOMAIN TRANSFORMATION

The domain transformation stage aims to convert tabular data into the desired form of images (*3, 84, 84*), while preserving the characteristics and semantics of the tabular data. We hypothesize that the difference between images and tabular data lies in the association with neighboring values and spatial relations(Zhu et al., 2021). The pixels in an image exhibit strong correlations with adjacent pixels, and this is why the kernels in a Convolutional Neural Network (CNN) play an important role. Therefore, we incorporate spatial relations into tabular data and undergo a process of shaping it into the desired form. Given $n$ features, we measure the Euclidean distance between these features and rank them to create an $(n, n)$ feature matrix, denoted as $\mathbf{R}$. Assume we have data matrix $\mathbf{D}$ with $C$ data samples and $n$ features($D_{ij}$ means the $j$ th feature of the $i$ th data sample.) and an array of feature name $F$. $F_i$ indicates the vector of the $i$ th feature name obtained by GloVe100(Pennington

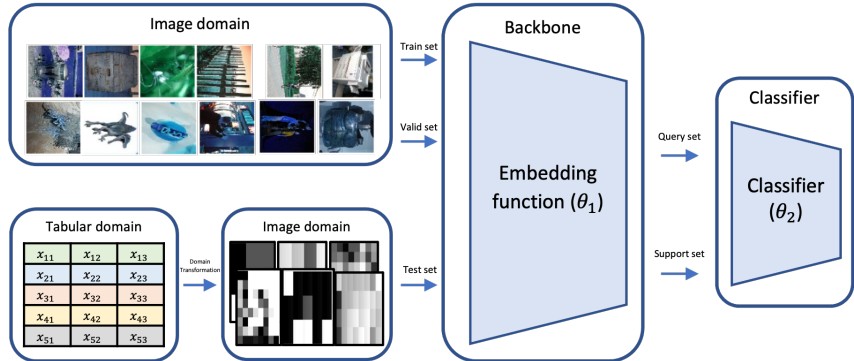

Figure 1: Overview of TablEye. The natural images of image domain are part of mini-ImageNet.

et al., 2014). If no meaningful name exists for a specific feature, we used '$i$ feature' as the feature name.

$$R_{ij} = \frac{1}{C} \sum_{c=0}^{C} \sqrt{(D_{ci} - D_{cj})^2} + \alpha \times \sqrt{(F_i - F_j)^2} \quad \text{where} \quad 0 < i \leq n \quad \text{and} \quad 0 < j \leq n$$

We also measure the distance and rank between $n$ elements to generate an $(n, n)$ pixel matrix, denoted as $\mathbf{Q}$. The pixel matrix $\mathbf{Q}$ is the similarity matrix between the coordinate pixels of $n_r \times n_c$ image. ($n = n_r \times n_c$ and $n_r$ and $n_c$ are the height and width of the transformed image.) Assume a coordinate list of $n$ features.

$$\text{Coordinates} = [(0, 0) \ldots (0, n_c - 1) \ldots (1, 0) \ldots (1, n_c - 1) \ldots (n_r - 1, 0) \ldots (n_r - 1, n_c - 1)]$$

The $i$ th element of the coordinate indicates the coordinate of $i$ th the feature for $N(= N_r \times N_c)$ image.

$$Q_{ij} = \sqrt{(\text{Coordinate}[i][0] - \text{Coordinate}[j][0])^2 + (\text{Coordinate}[i][1] - \text{Coordinate}[j][1])^2}$$

Then, we compute the Euclidean distance between $\mathbf{R}$ and $\mathbf{Q}$ and rearrange the positions of the features to minimize the distance, aiming to align the feature distance and pixel distance, thus assigning spatial relations. This results in obtaining a 2-dimensional image $M$ of size $n_r \times n_c$, where features with closer distances correspond to pixels that are closer to each other.

In the equations below, $r_{ij}$ and $q_{ij}$ represent the elements at the $i$-th row and $j$-th column of $\mathbf{R}$ and $\mathbf{Q}$, respectively. By minimizing the distance between $\mathbf{R}$ and $\mathbf{Q}$ according to the equations, we align the feature distance and pixel distance, thus assigning spatial relations.

$$-Loss(R, Q) = \sum_{i=1}^{N} \sum_{j=1}^{N} (r_{ij} - q_{ij})^2 \tag{1}$$

By repeating the same elements in a matrix $M$ of size $n_r \times n_c$, we obtain an image of size (84, 84). Applying the same (84, 84) image to each channel, we obtain an image of size (3, 84, 84). We refer to this image transformed from tabular data as the **tabular image**. Figure 2 represents the results of transforming one data sample from each of the six datasets(Vanschoren et al., 2014) used in the experiment into tabular images according to the proposed domain transformation method. Algorithm 1 at Appendix D shows the detailed process of domain transformation

### 3.3 LEARNING PRIOR KNOWLEDGE

The proposed TablEye model consists of a backbone that serves as an embedding function to operate in the suitable dimension for few-shot learning, and a classifier that performs the few-shot learning

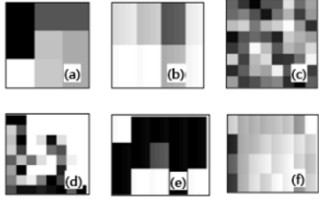

Figure 2: Example tabular images. (a), (b), (c), (d), (e) and (f) are tabular images from CMC, Diabetes, Karhunen, Optdigits, Lung and Cancer data respectively.

task based on the embedded support set. TablEye utilizes mini-ImageNet(Vinyals et al., 2016) to train backbone and classifier We adopted four different backbone architectures as shown in Figure 3. It is because the structure and training state of the backbone can significantly impact the training state of the classifier. Figure 3 illustrates the actual architectures of the four backbones, namely Resnet12, Conv2, Conv3, and Conv4, proposed and experimentally validated in this paper. The schematic diagram depicting the ResNet12 architecture is derived from the seminal work presented in the Choi et al. (2018) paper. Hereinafter, Resnet12, Conv2, Conv3, and Conv4 refer to each backbone depicted in Figure 3 within this paper. Resnet12 is a complex and deep backbone with a 12-layer ResNet(He et al., 2016) structure. Conv2, Conv3, and Conv4 are intuitive and shallow backbone architectures with 2, 3, and 4-layer CNN networks, respectively.

The backbone continuously learns to achieve a better embedding function for the classifier based on the predictions of the classifier using cross-entropy loss. The classifier plays a direct role in the few-shot learning task based on the embedded tabular images as latent vectors. In this paper, we adopt the principles of Prototypical Network(Snell et al., 2017), prototypes and inner loop adaptation of MAML(Finn et al., 2017) as our classifier. Both principles can be applied to various machine learning model structures. Moreover, recent studies have shown that few-shot learning with Prototypical Network achieves better performance than other complex few-shot learning models. Considering our goal of creating a model that operates with limited computing resources, we choose these two options for the classifier. When selecting the Proto-layer as the classifier, the classifier forms prototypes by averaging the latent vectors of the support sets for each class. It predicts the result by measuring the distances between the latent vectors of the query set and each prototype to determine the closest class. Alternatively, when selecting the MAML-layer as the classifier, we iteratively train a trainable fully connected layer within the inner loop using the latent vectors of the support set. The fully connected layer is then applied to the latent vectors of the query set to make predictions. Algorithm 2, 3 at Appendix D explain the detailed process of backbone and classifier.

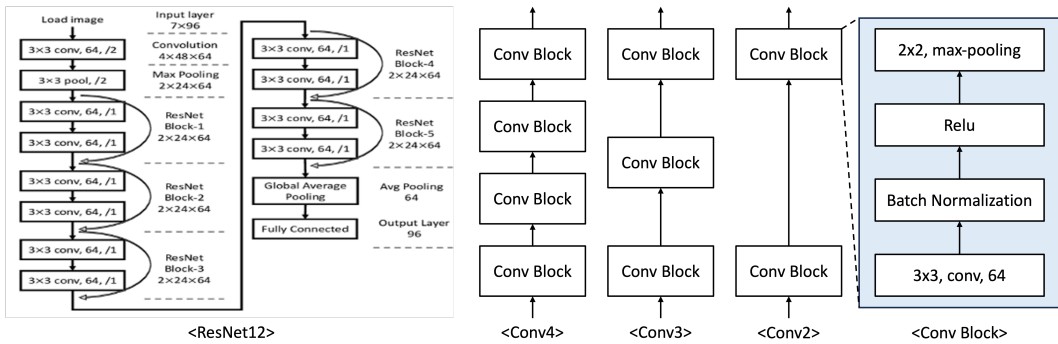

Figure 3: Four Backbone Structures of TablEye. Conv2, Conv3, Conv4 are composed of multiple Conv Block.

## 4 EXPERIMENTS

### 4.1 EXPERIMENTAL ENVIRONMENT

**Data** To validate the hypothesis of this paper, we conducted experiments using image data from mini-ImageNet(Vinyals et al., 2016) and open tabular data from OpenML(Vanschoren et al., 2014) and Kaggle. We constructed a train set consisting of 50,400 images and a validation set of 9,600 images from mini-ImageNet. For the test set, we composed tabular images after domain transformation. To ensure clear validation of the hypothesis, we applied the following criteria in selecting the tabular datasets for experiments: (1) Diversity of features: dataset containing only categorical features, dataset containing only numerical features, and dataset containing both categorical and numerical features, (2) Diversity of tasks: binary classification and multiclass classification, (3) Inclusion of medical data for industrial value validation. Appendix B shows the detail of the datasets.

**Notation** The abbreviation 'T-A-B ' signifies a condensed form of 'TablEye-A-B ', denoting the implementation of TablEye with 'A ' serving as the classifier and 'B ' as the backbone. Here, 'P ' and 'M ' denotes 'Proto-layer ' and 'MAML-layer ' . 'C2 ' 'C3 ' 'C4 ' and 'R ' represents 'Conv2 ', 'Conv3 ', 'Conv4 ' and 'Resnet12 '.

### 4.2 ABLATION STUDY

Throughout the research process, the main question was whether the prior knowledge learned from natural images could be applied to tabular images. To address this, we employed t-SNE(t-Distributed Stochastic Neighbor Embedding)(Van der Maaten & Hinton, 2008) technique to embed and visualize the distributions of natural images and transformed tabular images in a 2-dimensional space. Figure 4 visually presents the results of embedding into a two-dimensional space using t-SNE. Based on the 2-dimensional embedding results, we measured the maximum distance, denoted as $distance_{max}$, from the mean vector of natural images as the center of two circles, $c_1$ and $c_2$. We then drew two circles: circle $c_1$ with a radius of $distance_{max}$ and circle $c_2$ with a radius of 0.8 * $distance_{max}$. The scattered points in Figure 4 represent individual data samples, while the red and blue circles represent $c_1$ and $c_2$, respectively. We observed that some tabular images fell within $c_2$, while the majority of tabular images fell within $c_1$. Therefore, we concluded that there is no domain shift issue in learning the prior knowledge of tabular images from natural images.

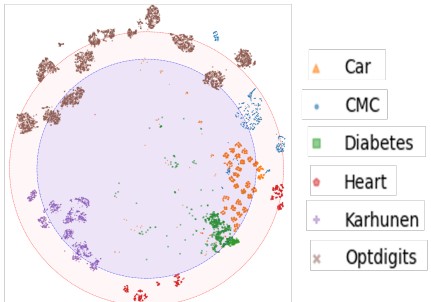

Figure 4: Visualization of Natural Image and Tabular Image Using T-SNE. Each points indicate tabular image, red circle(larger cirlce) indicates $c_1$ and blue circle(smaller circle) indicates $c_2$. The distinction of the six tabular datasets can be accomplished through the observation of the colors and shapes of the points.

To empirically substantiate the influence of acquiring prior knowledge from the image domain, we evaluated the accuracy of few-shot tabular classification under two different conditions: 1) directly applying few-shot learning algorithms designed for image data to tabular images, and 2) leveraging the mini-ImageNet dataset for prior knowledge acquisition before employing the same algorithms. When directly applying few-shot learning algorithms, we used randomly initialized backbone Table 1 elucidates the ramifications of incorporating prior knowledge from the image domain on the efficacy of few-shot tabular classification tasks. Excluding 1-shot scenarios for the accuracy of the T-P-R(CMC) and T-P-C3(Karhunen), we observed a substantial enhancement in performance in all

other cases when learning originated in the image domain. Thus, we have ascertained that the potency of TablEye not only stems from the few-shot learning algorithms but also from the benefits accrued through prior knowledge acquisition in the image domain.

Table 1: Comparison of Few-shot Tabular Classification Accuracy Based on Prior Knowledge Learning in the Image Domain. 'No Img' represents the condition where no prior knowledge learning has occurred in the image domain. Randomly initialized backbone is applied and trained on a tabular image. 'Img' denotes cases where prior knowledge has been acquired using mini-ImageNet. We report the mean of over 100 iterations.

|        |        | CMC | | Diabetes | | Karhunen | | Optdigits | |
|--------|--------|--------|-------|--------|-------|--------|-------|--------|-------|
|        |        | No Img | Img | No img | Img | No img | Img | No img | Img |
| 1-shot | T-P-R  | 35.36 | 35.97 | 54.67 | 58.83 | 30.57 | 30.50 | 43.11 | 44.32 |
|        | T-P-C2 | 36.42 | 37.33 | 56.26 | 56.53 | 45.39 | 51.21 | 64.58 | 71.18 |
|        | T-P-C3 | 37.46 | 37.31 | 55.68 | 57.43 | 44.42 | 51.39 | 63.59 | 70.30 |
|        | T-P-C4 | 36.84 | 37.45 | 54.88 | 57.79 | 41.67 | 44.85 | 62.96 | 65.76 |
| 5-shot | T-P-R  | 36.35 | 38.37 | 56.48 | 64.39 | 31.80 | 41.18 | 41.77 | 51.83 |
|        | T-P-C2 | 38.99 | 40.34 | 57.65 | 65.15 | 41.78 | 77.94 | 62.54 | 87.44 |
|        | T-P-C3 | 38.50 | 41.22 | 57.23 | 66.20 | 40.98 | 74.61 | 62.02 | 86.83 |
|        | T-P-C4 | 38.27 | 40.89 | 55.59 | 68.73 | 37.86 | 70.72 | 61.54 | 84.58 |

## 4.3 COMPARISON RESULTS WITH TABLLM

**Data** The dataset for TabLLM(Hegselmann et al., 2023) is constrained by token length and the absence of meaningful feature names, which restricts its applicability to datasets such as Karhunen and Optdigits. The datasets utilized in the other experiments, Karhunen and Optdigits, comprised 65 features, rendering TabLLM experiments infeasible. Moreover, these datasets lacked meaningful feature names. Consequently, alternative datasets used in experiments of previous work were selected to replace those. The Diabetes dataset exclusively comprises numerical features, the Heart dataset encompasses both numerical and categorical features, and the Car dataset solely comprises categorical features.

**Metric** For the Metric, we used the AUC (Area Under the Receiver Operating Characteristic Curve) metric to compare our method under the same conditions as TabLLM.

**Shot setting** In the paper of TabLLM, comparisons were made from 4-shot to 512-shot. We assume, however, a few-shot scenario, we compared the AUC under 4-shot, 8-shot, and 16-shot conditions.

TabLLM transforms tabular data consisting of categorical and numerical features into prompts that can be understood by language model. It leverages the prior knowledge of language models using these prompts. Table 2 displays the performance comparison between our approach, table-to-image and TabLLM, table-to-text method. TablEye exhibited superior performance to previous work in numeric-only datasets, Diabetes and Heart and showed similar or superior performance in the Categoric-only dataset, Car. TabLLM showed best performance in 4-shot scenarios but T-M-C4 demonstrated 0.89 AUC that was 0.03 higher than TabLLM in 16-shot scenarios.

Tableye exhibited an approximately 0.1 higher AUC than TabLLM on the diabetes dataset. We believe this is due to TabLLM's power diminishing in numeric-only datasets, which are more distant from general language. However, in 4-shot scenarios of car datasets, TablEye consistently showed lower performance compared to table-to-text method. We speculate that this is because of the nature of TabLLM utilizing language model, better understands categorical features.

TabLLM has approximately 11 billion parameters((Sanh et al., 2021)), while TablEye utilizes up to 11 million parameters ResNet12 exhibits parameters that are approximately 1/916 the size of TabLLM. Conv2, Conv3, and Conv4 display parameter sizes that span a range from 1/97,345 to 1/282,051 when compared to TabLLM. TablEye has a significantly smaller model size compared to the table-to-text method. Our approach also can demonstrate comparable or superior performance and extremely efficient computation power. Appendix C provides the detailed information.

Table 2: Few Shot Tabular Classification Test AUC performance on 3 tabular datasets. We used the AUC performance of XGB, TabNet, SAINT and TabLLM from TabLLM paper. The bold indicates result within 0.01 from highest accuracy.

|  | Diabetes | | | Heart | | | Car | | |
|---|---|---|---|---|---|---|---|---|---|
|  | 4-shot | 8-shot | 16-shot | 4-shot | 8-shot | 16-shot | 4-shot | 8-shot | 16-shot |
| XGB | 0.50 | 0.59 | 0.72 | 0.50 | 0.55 | 0.84 | 0.50 | 0.59 | 0.70 |
| TabNet | 0.56 | 0.56 | 0.64 | 0.56 | 0.70 | 0.73 | ** | 0.54 | 0.64 |
| SAINT | 0.46 | 0.65 | 0.73 | 0.80 | **0.83** | **0.88** | 0.56 | 0.64 | 0.76 |
| TabLLM | 0.61 | 0.63 | 0.69 | 0.76 | **0.83** | **0.87** | **0.83** | **0.85** | 0.86 |
| T-P-R | 0.68 | 0.70 | 0.69 | 0.72 | 0.78 | 0.69 | 0.69 | 0.68 | 0.75 |
| T-P-C2 | 0.68 | 0.68 | 0.68 | 0.84 | **0.83** | 0.85 | 0.79 | 0.79 | 0.79 |
| T-P-C3 | **0.71** | **0.73** | 0.71 | **0.86** | 0.79 | 0.78 | 0.72 | 0.71 | 0.76 |
| T-P-C4 | **0.72** | 0.71 | 0.69 | 0.82 | 0.81 | 0.79 | 0.79 | 0.83 | 0.83 |
| T-M-C2 | 0.68 | **0.73** | **0.78** | 0.81 | **0.83** | 0.82 | 0.74 | 0.82 | 0.86 |
| T-M-C3 | **0.71** | **0.74** | 0.76 | 0.73 | **0.83** | 0.83 | 0.78 | **0.85** | 0.87 |
| T-M-C4 | 0.69 | **0.74** | 0.75 | 0.82 | **0.84** | **0.88** | 0.75 | 0.82 | **0.89** |

## 4.4 FEW-SHOT CLASSIFICATION RESULTS WITH BASELINE

**Baseline** We chose a supervised learning models that can be experimented within a meta-learning setting without an unlabeled set. We selected both tree-based model and neural network-based model known for their high performance about tabular learning (Shwartz-Ziv & Armon, 2022).

**STUNT** A fixed number of unlabeled sets were used as the train set. For the CMC, Diabetes, Karhunen, and Optdigits datasets, 441, 230, 600, and 1686 unlabeled sets were respectively utilized.

Table 3 displays the performance of TablEye. The results demonstrate the superiority of TablEye over traditional methods such as XGB and TabNet(Arik & Pfister, 2021), and even over STUNT, which is state of the art about few-shot tabular learning. In the 1-shot setting, methods of Tabl-Eye, T-P-C2 and T-P-C3 exhibited the highest average accuracies of 54.06% and 54.11%, respectively, outperforming all other methods. The performance advantage of TablEye was also evident in the 5-shot setting, where the T-P-C2 and T-P-C3 methods continued to outperform other methods, achieving average accuracies of 67.72% and 67.22%, respectively.

STUNT(Nam et al., 2023) showed a considerable performance with average accuracies of 50.94% and 66.46% in the 1-shot and 5-shot settings respectively. The performance of STUNT is, however, heavily influenced by the size of the unlabeled dataset. In real-world industrial processes, obtaining a sufficiently large and well-composed unlabeled dataset is often challenging, making superior performance of TablEye without relying on unlabeled data highly notable.

## 5 DISCUSSION

TabLLM was unable to handle datasets with more than a certain number of features or meaningless feature names, such as the Karhunen and Optdigits datasets with 65 features and feature names like f1, f2, and f3. It is because of the limitations in the token size of the LLM and the necessity for meaningful feature names. The results of our approach confirmed higher performance compared to table-to-text method, particularly in datasets with numerical features such as Diabetes and Heart. Comparing the size of the TabLLM and TablEye previous work possessed a significantly larger number of parameters, requiring considerably higher computational power. Nevertheless, our method demonstrated superior performance with the Diabetes and Heart datasets. Thus, we conclude that our approach is more efficient and showed similar or superior performance for various datasets, overcoming the limitations of TabLLM, which has restrictions on the datasets it can handle and requires high computational power.

STUNT requires a substantial amount of unlabeled data for training. The model used 80% of the total data as an unlabeled set for training in its paper. In this study, we aimed to use as little unlabeled data as possible to conduct experiments under similar conditions to other baselines, utilizing

Table 3: Few Shot Classification test accuracy(%) on 4 public tabular dataset. We report the mean of over 100 iterations. The bold indicates result within 1% from highest accuracy.

|        | Method | CMC | Diabetes | Karhunen | Optdigits | Average |
|--------|--------|-----|----------|----------|-----------|---------|
| 1-shot | XGB | 33.33 | 50.00 | 20.00 | 20.00 | 30.83 |
|        | TabNet | 34.84 | 51.90 | 21.97 | 20.45 | 32.29 |
|        | STUNT | 36.52 | 51.60 | 47.72 | 67.92 | 50.94 |
|        | T-P-R | 35.97 | **58.83** | 30.50 | 44.32 | 42.41 |
|        | T-P-C2 | **37.33** | 56.53 | **51.21** | **71.18** | **54.06** |
|        | T-P-C3 | **37.31** | 57.43 | **51.39** | **70.30** | **54.11** |
|        | T-P-C4 | **37.45** | 57.79 | 44.85 | 65.76 | 51.46 |
|        | T-M-C2 | 36.60 | **58.34** | 41.92 | 62.04 | 49.73 |
|        | T-M-C3 | **37.26** | **58.57** | 43.27 | 60.18 | 49.82 |
|        | T-M-C4 | **37.30** | 57.30 | 43.45 | 60.53 | 49.65 |
| 5-shot | XGB | **42.18** | 61.20 | 68.21 | 73.19 | 61.19 |
|        | TabNet | 36.07 | 50.23 | 20.28 | 21.33 | 31.98 |
|        | STUNT | **41.36** | 55.43 | **83.00** | 86.05 | 66.46 |
|        | T-P-R | 38.37 | 64.39 | 41.18 | 51.83 | 48.94 |
|        | T-P-C2 | 40.34 | 65.15 | 77.94 | **87.44** | **67.72** |
|        | T-P-C3 | **41.22** | 66.20 | 74.61 | **86.83** | **67.22** |
|        | T-P-C4 | 40.89 | **68.73** | 70.72 | 84.58 | 66.23 |
|        | T-M-C2 | 37.65 | 63.18 | 56.38 | 62.79 | 55.00 |
|        | T-M-C3 | 38.48 | 64.35 | 44.80 | 58.79 | 51.60 |
|        | T-M-C4 | 37.95 | 65.94 | 59.12 | 71.85 | 58.71 |

approximately 30% of the data as the unlabeled set for training. Despite employing a considerable number of unlabeled sets in experiments, TablEye, which did not use any unlabeled sets, showed higher accuracy than STUNT. Therefore, we believe that TablEye has overcome the limitations of STUNT, which requires a large unlabeled set.

When applying TablEye to medical datasets, we observed markedly higher accuracy compared to other baselines. Compared to existing methods, we achieved an average accuracy of 15% higher in 1-shot scenarios and approximately 2% higher in 5-shot scenarios. These results indicate that our method can produce meaningful results not only in public tabular data but also in medical data of industrial value.

## 6  CONCLUSION

In this paper, we propose TablEye, a novel few-shot tabular learning framework that leverages prior knowledge acquired from the image domain. TablEye performs a transformation of tabular data into the image domain. It then utilizes prior knowledge gained from extensive labeled image data to execute few-shot learning. Our experiments on various public tabular datasets affirm the efficacy of TablEye. Experimental results indicate a notable increase in performance metrics; TablEye surpasses TabLLM by a maximum of 0.11 AUC except for one 4-shot learning and demonstrates an average accuracy enhancement of 3.17% over STUNT in the 1-shot learning scenario. Notably, our approach effectively overcomes several limitations including a dependence on the number and names of features in the dataset, the need for substantial computational power, and the requirement for a large unlabeled set. We believe that leveraging the image domain to solve problems in the tabular domain opens up exciting new possibilities for advancing the field of tabular learning.

## 7  REPRODUCIBLITY

To reproduce the framework proposed in this paper, the few-shot learning task process was implemented using the LibFewShot library[36]. A configuration file for utilizing the LibFewShot library and detailed model setting for reproduction will be made publicly available on GitHub.

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

## A   FEW-SHOT LEARNING BACKGROUND

The Few-shot Learning problem is a type of machine learning problem characterized by having a limited amount of labeled training data. Equations (1), (2), (3) show the relation. It is particularly relevant in situations where the data set containing the train set $D_{\text{train}}$ and the test set $D_{\text{test}}$, $D_{\text{train}}$ containing the sample pairs $(x_i, y_i)$ with $i$ ranging from 1 to small size $I$, and $D_{\text{test}}$ consisting of $x^{\text{test}}$. Applications of few-shot learning include image classification, sentiment classification, and object recognition. The general notation of $N$-way-$K$-shot classification refers to a train set composed of $N$ classes, each with $K$ examples, totaling $I = NK$ examples.

$$D = \{D_{\text{train}}, D_{\text{test}}\} \tag{1}$$

$$D_{\text{train}} = \{(x_i, y_i)\}_{i=1}^I \tag{2}$$

$$D_{\text{test}} = \{x^{\text{test}}\} \tag{3}$$

The objective of solving the few-shot learning problem is to find an optimal hypothesis $\hat{h}$, such that $\hat{y} = \hat{h}(x; \theta)$, in the hypothesis space $H$ ($\hat{h} \in H$). This can be a challenging task due to the large hypothesis space $H$ and the small size of the labeled set $I$, which makes traditional supervised learning techniques, which require a large amount of labeled data, unfeasible. Equations (4), (5) show the relationship of expected risk $R$ and empirical risk $R_I$ with $x$, $y$(Nathani & Singh, 2021; Vapnik, 1991). To minimize the expected risk $R$, we approximate it using the empirical risk $R_I$, since the true underlying probability distribution $p(x, y)$ is unknown.

$$R(h) = \int L(h(x), y) \, dp(x, y) \tag{4}$$

$$R_I(h) = \frac{1}{I} \sum_{i=1}^I L(h(x_i), y_i) \tag{5}$$

However, finding the optimal hypothesis $\hat{h}$ remains elusive. To address this challenge, we decompose the total error of the problem by using the best approximation $h^*$ for $\hat{h}$ as follows(Bottou & Bousquet, 2007; Bottou et al., 2018). Equation (6) shows the decomposition of total error. $H$ is the size of hypothesis space, and $I$ is the size of the training set.

$$E[R(h_I) - R(h^*)] = E[R(h^*) - R(\hat{h})] + E[R(h_I) - R(h^*)]$$

$$= \epsilon_{\text{approximation}}(H) + \epsilon_{\text{estimation}}(H, I) \tag{6}$$

According to the formula (6), the total error is comprised of the approximation error, $\epsilon_{\text{approximation}}$ of the hypothesis space $H$ and the estimation error, $\epsilon_{\text{estimation}}$ of our hypothesis $h^I$ with the best approximation $h^*$. The formula highlights that the size of the hypothesis space $H$ and the size of the training set $I$ both play a critical role in the few-shot learning problem.

The hypothesis space and the size of the training set are crucial factors in few-shot learning. As a result, there are generally three approaches to solving few-shot learning. First, increasing the size of the training set $I$. Second, defining the hypothesis space $H$ from the perspective of the model. Third, changing the search strategy for the best approximation $h^*$ within the hypothesis space from the perspective of the algorithm. The approach from the perspective of data involves augmenting the small training set by leveraging prior knowledge, such as through transforming the data samples from the training set, weakly labeled set, unlabeled set, or similar datasets. The approach from the perspective of the model reduces the size of the hypothesis space by utilizing prior knowledge through multi-task learning or embedding learning. Lastly, the approach from the perspective of the algorithm adjusts the strategy for searching for the optimal hypothesis $\hat{h}$ within the hypothesis space $H$, such as by refining parameters or learning the optimizer.

## B    DATASET DESCRIPTION

Table 4 shows a detailed composition of dataset used for experiments. The # indicates the number. The column named 'Type' indicates the type of feature including numeric only feature, categoric only feature and numeric and categoric feature. The N and C of 'Type' indicate numeric and categoric only feature. The M of 'type' indicates numeric and categoric mixed feature. The diabetes, Lung, Cancer, and Prostate datasets are medical datasets used for Table 7.

Table 4: Description of Dataset

| Dataset | Size | # of features | # of classes | Type |
|---------|------|---------------|--------------|------|
| CMC | 1473 | 10 | 3 | M |
| Diabetes | 768 | 9 | 2 | N |
| Karhunen | 2000 | 65 | 10 | N |
| Optdigits | 5620 | 65 | 10 | N |
| Car | 1728 | 7 | 4 | C |
| Heart | 918 | 12 | 2 | N |
| Lung | 309 | 16 | 2 | M |
| Cancer | 569 | 32 | 2 | N |
| Prostate | 100 | 9 | 2 | N |

## C    NUMBER OF PARAMETERS

Table 5 shows detailed information of the model size. You can see the large difference in model size between TabLLM and TablEye. The number of TabLLM parameters is up to about 97345 times larger than TablEye.

Table 5: Model size of TabLLM and TablEye. The M-rate signifies the size of the model in comparison to TabLLM. It is calculated as the number of parameters divided by 11 billion.

| | TabLLM | T-P-R | T-P-C2 | T-P-C3 | T-P-C4 |
|---|---|---|---|---|---|
| # of Parameters | 11B | 12M | 39K | 76K | 113K |
| M-rate | 1 | 1/916.7 | 1/282051.3 | 1/144736.8 | 1/97345.1 |

| | TabLLM | T-M-C2 | T-M-C3 | T-M-C4 |
|---|---|---|---|---|
| # of Parameters | 11B | 47K | 84K | 121K |
| M-rate | 1 | 1/234042.5 | 1/130925.3 | 1/90909.0 |

## D    ALGORITHM

In this section, we describe the algorithms of the domain transformation and the two classifiers used in our experiments, the Proto-layer and MAML-layer using Pseudo code. Before performing the Domain Transformation Algorithm, min-max normalization was applied to each feature of the tabular data. Then, after Domain Transformation, the value is multiplied by 255.

## E    INPUT IMAGE CHANNEL SELECTION

Our backbone models operate by taking three-channel images as input. To create three-channel images for domain transformation, the same values are stacked to form three channels. This raises a question as to why we generate arbitrary three-channel images instead of treating all images as

---

**Algorithm 1** Domain Transformation

---

1: **Step1**: Generate feature distance matrix **R**.
2: We have data matrix **D** with $C$ data samples and $n$ features and an array of feature name $F$.
$\qquad R_{ij} = \frac{1}{C} \sum_{c=0}^{C} \sqrt{(D_{ci} - D_{cj})^2} + \alpha \times \sqrt{(F_i - F_j)^2}$   where   $0 < i \le n$   and   $0 < j \le n$
3: **Step2**: Generate pixel distance matrix **Q**.

$\qquad \text{Coordinate} = [(0,0) \dots (0, n_c - 1) \dots (1, 0) \dots (1, n_c - 1) \dots (n_r - 1, 0) \dots (n_r - 1, n_c - 1)]$

$\qquad Q_{ij} = \sqrt{(\text{Coordinate}[i][0] - \text{Coordinate}[j][0])^2 + (\text{Coordinate}[i][1] - \text{Coordinate}[j][1])^2}$

4: **Step3**: Minimize the error between the **Q** and **R**
5: By repeating swapping, we minimize the error, the distance between Q and R. It makes the positional coordinate of the tabular image more like feature similarity. Make the distance of features like pixels natural.
6: $F = [F_1, F_2, F_3, \dots F_n]$
7: **for** `max iteration` **do**
8:     Select the latest updated feature $F_l$
9:     Compute the error of swapping $F_l$ with other feature $F_s$.
10:     Select the feature $F_s = \arg\max(\text{err}(R, Q) - \text{err}(R^*, Q))$     $\triangleright R^*$ is the swapped and regenerated feature matrix.
11:     **if** $\text{err}(R, Q) - \text{err}(R^*, Q) >$ standard **then**
12:         Swap $F_l$ and $F_s$ in $F$ and regenerate feature matrix **R**.
13:     **else**
14:         `Patience` $- = 1$
15:     **end if**
16: **end for**
17: **if** `patience` is 0 **then**
18:     Make tabular data into 2D matrix with the order of R.
19:     **for** $i$ in $0, 1, 2, \dots, N$ **do**
20:         2D matrix$[i//N_c][i\%N_r] = F[i]$
21:     **end for**
22:     Repeat the element of matrix with size of $(84, 84)$
23:     Stack three channels to make a $(3, 84, 84)$ image
24: **end if**

---

**Algorithm 2** Proto-layer

---

1: **(Proto-layer)**
2: **function** TRAIN
3:     $Image \leftarrow$ BACKBONE(all images)
4:     support images, support targets, query images, query targets $\leftarrow batch$
5:     $Output \leftarrow$ CLASSIFIER(support images, query images)     $\triangleright$ Classifier computes the prototype using support images. No trainable parameter.    $\triangleright$ Output denotes the class to which the query images belong.
6:     $Loss \leftarrow$ CROSSENTROPYLOSS($Output$, query targets)
7:     LOSS.BACKWARD
8: **end function**

---

**Algorithm 3** MAML-layer

---

1: Classifier is a 1-layer fc (fully connected layer)
2: **function** TRAIN
3:     support images, support targets, query images, query targets $\leftarrow batch$
4:     Initialize an empty list: $output\_list \leftarrow []$
5:     **for** $i$ in 1 **to** iteration **do**
6:         Perform fast adaptation on the $i$-th support image and target
7:         $output \leftarrow$ classifier(backbone(query image$[i]$))
8:         Append $output$ to $output\_list$
9:     **end for**
10:     $output \leftarrow$ concatenate all outputs in $output\_list$
11:     $loss \leftarrow$ cross-entropy loss of $output$ and query targets
12:     Perform backpropagation on $loss$
13: **end function**

---

grayscale. This is because using three-channel images as inputs introduces overhead at the input layer. Table 6 resolves this question by showing the results of experiments treating all images as three-channel and comparing them to experiments treating images as one-channel, grayscale images. In practice, the use of grayscale images in some cases yields performance almost equivalent to, or slightly higher than, three-channel images. However, in most cases, three-channel images demonstrate superior performance. We hypothesize that this is because the backbone can learn richer prior knowledge when dealing with three-channel images. However, the precise reasons for this require further research.

Table 6: Impact of Input Image Channel. 1-ch indicates 1-channel input image and 3-ch indicate 3-channel input image.

|  |  | CMC | | Diabetes | | Karhunen | | Optidigits | |
|---|---|---|---|---|---|---|---|---|---|
|  |  | 1-ch | 3-ch | 1-ch | 3-ch | 1-ch | 3-ch | 1-ch | 3-ch |
| 1-shot | T-P-C2 | 36.52 | 37.33 | 56.83 | 56.53 | 42.91 | 51.21 | 59.99 | 71.18 |
|  | T-P-C3 | 36.44 | 37.31 | 57.76 | 57.43 | 49.16 | 51.39 | 68.23 | 70.30 |
|  | T-P-C4 | 36.78 | 37.45 | 56.42 | 57.79 | 45.91 | 44.85 | 65.91 | 65.76 |
| 5-shot | T-P-C2 | 40.72 | 40.34 | 63.59 | 65.15 | 64.80 | 77.94 | 83.68 | 87.44 |
|  | T-P-C3 | 40.98 | 41.22 | 62.77 | 66.20 | 72.88 | 74.61 | 82.99 | 86.83 |
|  | T-P-C4 | 40.28 | 40.89 | 62.92 | 68.73 | 73.51 | 70.72 | 85.10 | 84.58 |

## F  MEDICAL DATASETS FEW-SHOT RESULTS

**Data** We obtained medical datasets on diabetes and three types of cancer, which were publicly available on Kaggle. We aim to validate the applicability of our method to real medical datasets.

Table 7 illustrates the accuracy of the few-shot tabular binary classification test on four public tabular medical datasets. These results affirm the enhanced performance of our novel few-shot tabular learning framework in medical data classification tasks, surpassing conventional methods. The significant increase in average accuracy for our methods in both 1-shot and 5-shot scenarios underscores the efficacy of our approach. Specifically, T-P-C4 and T-M-C4, exhibiting superior performance when trained with more data (in a 5-shot setting), further underscores the effectiveness of our method in few-shot learning scenarios.

Table 7: Few Shot Binary Classification test accuracy(%) on 4 public tabular medical datasets. We report the mean of over 100 iterations. The bold indicates result within 1% from highest accuracy.

|  | Method | Diabetes | Lung | Cancer | Prostate | Average |
|---|---|---|---|---|---|---|
| 1-shot | XGB | 50.00 | 50.00 | 50.00 | 50.00 | 50.00 |
|  | TabNet | 51.90 | 52.03 | 50.00 | 56.07 | 52.50 |
|  | T-P-C2 | 56.53 | 63.02 | **85.88** | 58.50 | 65.98 |
|  | T-P-C3 | **57.43** | 64.93 | **85.08** | **66.51** | **68.49** |
|  | T-P-C4 | **57.79** | **66.64** | **85.64** | 65.18 | **68.81** |
|  | T-M-C2 | 56.53 | 61.70 | 84.53 | 62.74 | 66.38 |
|  | T-M-C3 | **57.43** | 61.44 | 83.36 | 62.46 | 66.17 |
|  | T-M-C4 | **57.79** | 62.99 | 82.33 | 62.36 | 66.38 |
| 5-shot | XGB | 61.20 | 63.17 | 81.87 | **78.10** | 71.08 |
|  | TabNet | 50.23 | 50.43 | 50.17 | 58.57 | 52.35 |
|  | T-P-C2 | 65.15 | 70.43 | 86.90 | 62.75 | 71.31 |
|  | T-P-C3 | 66.20 | 65.62 | 87.02 | 63.97 | 70.70 |
|  | T-P-C4 | **68.73** | 66.69 | 88.92 | 66.98 | 72.83 |
|  | T-M-C2 | 65.15 | 73.34 | 91.91 | 73.42 | 75.96 |
|  | T-M-C3 | 66.20 | 67.85 | 91.51 | 74.49 | 75.01 |
|  | T-M-C4 | **68.73** | **72.80** | **92.10** | 74.34 | **76.96** |

## G    TABLE TO IMAGE CONVERSION METHOD

There are various methods for converting one-dimensional tabular data into two-dimensional image data. Table 8 demonstrates the performance differences of our used domain transformation method compared to other methods, including simple shape manipulation into two dimensions and visualization through graphs. The 'Direct' method in domain transformation involves altering the order of features, omitting the step of adding spatial relations. In contrast, the 'Graph' method plots tabular data as graphs. The Graph method, though significantly different from domain transformation, showed considerably high performance. However, the Direct method exhibited notably lower performance. This indicates that adding spatial relations contributes to meaningful performance improvements.

Table 8: Table to Image Conversion Method Comparison. Direct is just 2d shaping tabular data into tabular images without spatial relation. Graph is visualizing tabular data into a format of graph by plotting. Ours is the domain transformation method.

|  | Diabetes | | | Karhunen | | | Optdigits | | |
|---|---|---|---|---|---|---|---|---|---|
|  | Direct | Graph | Ours | Direct | Graph | Ours | Direct | Graph | Ours |
| 1-shot | 50.420 | 51.877 | 56.53 | 19.859 | 40.796 | 51.21 | 35.400 | 46.504 | 71.18 |
|  | 49.880 | 51.627 | 57.43 | 20.161 | 41.757 | 51.39 | 37.073 | 64.936 | 70.30 |
|  | 49.887 | 51.630 | 57.79 | 19.879 | 38.179 | 44.85 | 35.737 | 58.903 | 65.76 |
| 5-shot | 50.467 | 53.853 | 65.15 | 20.009 | 56.789 | 77.94 | 44.413 | 71.233 | 87.44 |
|  | 49.847 | 52.940 | 66.20 | 20.035 | 55.393 | 74.61 | 43.624 | 84.129 | 86.83 |
|  | 50.117 | 53.263 | 68.73 | 19.987 | 56.531 | 70.72 | 44.283 | 83.985 | 84.58 |

## H    POTENTIAL OF VISION TRANSFORMER BACKBONE

Table 9 shows the results of experiments conducted using the Vision Transformer (ViT)(Dosovitskiy et al., 2020) as the backbone and the Proto-layer as the classifier, performed under identical conditions. The results indicate that while ViT demonstrates respectable performance across all four datasets, it falls short of surpassing the other methods employed in TablEye. We believe this limitation is due to the fact that, unlike the other backbone-based models in TablEye that underwent extensive experimentation and optimization of learning parameters, ViT did not undergo this process. However, these experimental results are significant as they confirm the potential applicability of tabular image processing in the novel architecture of ViT.

Table 9: TablEye Accuracy leveraging ViT as backbone

|  |  | CMC | Diabetes | Karhunen | Optdigits |
|---|---|---|---|---|---|
| T-P-V | 1-shot | 35.93 | 56.78 | 43.79 | 67.27 |
|  | 5-shot | 40.1 | 62.65 | 69.76 | 64.13 |

