# OpenReview forum: "TABLEYE: SEEING SMALL TABLES THROUGH THE LENS OF IMAGES"
_ICLR.cc/2024/Conference — Submitted to ICLR 2024_

### Official Review · Reviewer_6PPe · 2023-10-27

**Soundness:** 3 good
**Presentation:** 3 good
**Contribution:** 2 fair
**Rating:** 5
**Confidence:** 3

**Summary:**

This paper proposes to transform tabular data into image formats and utilize pretrained vision models to help the learning of tabular few-shot learning. The experiment results show that the method can be better than a strong LLM-based method.

**Strengths:**

- Very simple and effective method.
- Transforming tabular data into image format is intuitive and novel.
- The proposed method has good performance even with a small visual encoder, better than an LLM-based method, which is very promising.

**Weaknesses:**

- Only two papers are discussed in the related work section, which makes reader difficult to place the paper in an appropriate context.
- The relationship between the quality of the visual encoder and the few-shot tabular performance is not shown.
- Missing an important baseline (See questions).
- Only the domain transformation module is proposed by the authors. Novelty is somewhat lacking.

**Questions:**

- Can you discuss more related works in the paper? For example, a brief introduction to the tabular learning literature.
- Can you give results using a more powerful visual encoder? In Luo et. al [1], it has been shown that better visual encoders can lead to better few-shot learning performance. Perhaps, you can try to use pretrained CLIP [2] or DINO-v2 [3] and report the results.
- Another straight way of transforming the tabular data into images is to directly visualize the table on an image in its original form. This should be a baseline to illustrate the advantage of your proposed tabular data transformation.

[1] A Closer Look at Few-shot Classification Again. ICML 2023.

[2] Learning Transferable Visual Models From Natural Language Supervision. ICML 2021.

[3] DINOv2: Learning Robust Visual Features without Supervision.

---

> ### Author Response · Authors · 2023-11-19
> **Rebuttal**
>
> **Q1. Can you discuss more related works in the paper? For example, a brief introduction to the tabular learning literature.**
>
> - Thank you for your question. I realize that I did not consider that readers might be unfamiliar with the concept of few-shot learning. For those who are completely new to few-shot learning, We have added a background section in Appendix A on page 12 of the paper to provide essential context and understanding.
>
>
> **Q2. Can you give results using a more powerful visual encoder? In Luo et. al [1], it has been shown that better visual encoders can lead to better few-shot learning performance. Perhaps, you can try to use pretrained CLIP [2] or DINO-v2 [3] and report the results.**
>
> *[1] A Closer Look at Few-shot Classification Again. ICML 2023.*
>
> *[2] Learning Transferable Visual Models From Natural Language Supervision. ICML 2021.*
>
> *[3] DINOv2: Learning Robust Visual Features without Supervision.*
>
> - We conducted experiments using CLIP's ViT-B/32 encoder as the backbone and the Proto-layer as the classifier. Although we anticipated the excellent few-shot performance of CLIP, the experimental results showed that CLIP did not learn effectively on any of the four datasets (CMC, Diabetes, Karhunen, Optdigits). We believe this is because CLIP's pre-trained visual encoder is designed to reduce the distance between the image and text domains, which is a different concept compared to backbones like conv2, conv3, which were trained specifically for natural image few-shot classification.
>
> - Therefore, we also conducted experiments using the ViT as the backbone and the Proto-layer as the classifier. The results showed that ViT performed reasonably well on all four datasets but did not surpass the performance of other methods used in TablEye. We attribute this to the fact that while the learning parameters of the current models have been optimized through numerous experiments, ViT has not undergone such optimization. However, this also indicates the potential applicability of the ViT structure to tabular images. We have added these experimental results to Appendix H on page 16 of the paper.
>
>
> **Q3. Another straight way of transforming the tabular data into images is to directly visualize the table on an image in its original form. This should be a baseline to illustrate the advantage of your proposed tabular data transformation.**
>
> - Thank you for mentioning an important baseline. We conducted additional experiments with two methods, including the baseline you mentioned, and compared them with domain transformation. The graph below experimentally demonstrates that domain transformation more effectively converts tabular data into a form suitable for CNNs compared to simply changing the order and size of tabular data to form a 2D image and representing it graphically. We have added a section in Appendix G on page 13, 15 of the paper to provide this experiment. Thank you for providing important comments that helped improve the paper.

---

> > ### Comment · Reviewer_6PPe · 2023-11-21
> >
> > Thanks for your response. While I am satisfied with the response to Q3, I still have some concerns about Q1 and Q2.
> >
> > 1. Note that what I mean by "context" is related work, not problem setup. The authors should discuss more related work relevant to few-shot learning, tabular learning, and few-shot tabular learning, instead of introducing the setup in the appendix.
> >
> > 2. (minor) Since I have plenty of experience in visual few-shot learning, I deeply doubt the experiment with CLIP. As evidenced by Luo et. al [1], the visual encoder of CLIP performs extremely well on each of the diverse 10 datasets (including several natural image datasets), thus there is no reason that CLIP cannot perform better than a miniImageNet-trained model on table-based images. Besides, I do not see any value in replacing the backbones with ViT. What I mean is to use better pretrained models trained on massive datasets to see the relationship between the **quality** of the visual encoder and the few-shot tabular performance. Note that this is not a major problem to reject the paper, but doing this can better enhance the value of the paper.

---

> > > ### Author Response · Authors · 2023-11-22
> > > **Rebuttal**
> > >
> > > **Q1. Note that what I mean by "context" is related work, not problem setup. The authors should discuss more related work relevant to few-shot learning, tabular learning, and few-shot tabular learning, instead of introducing the setup in the appendix.**
> > >
> > > - It seems we misunderstood your question earlier. We have revised the content of the related work section to reflect your opinion. The following content has been added to the related work section, providing a general explanation of tabular learning, few-shot learning, and few-shot tabular learning. Additionally, we have adjusted the descriptions of STUNT and TabLLM in the related work.
> > >
> > > *→  Tabular learning refers to the process of learning the mapping between input and output data using tabular data[1].
> > > Tabular data is often also called structured data[2] and is a subset of heterogeneous data presented in a table format with rows and columns. Each feature in this data is composed of either categorical or numerical features.
> > > Currently, methods based on decision trees and those based on Multi-Layer Perceptrons (MLP) are showing almost equal performance.
> > > Tabular learning still requires a large amount of labeled data.
> > > In the image domain, few-shot learning can easily acquire prior knowledge using many related images. For example, ProtoNet (Prototypical Network)\citep{snell2017prototypical} learns using similarities between images, and MAML (Model-Agnostic Meta-Learning)\citep{finn2017model} quickly adjusts the model across various tasks, enabling rapid learning with limited data.
> > > However, in the tabular domain, there are no equivalent sets of related tabular data. Therefore, few-shot tabular learning faces significant challenges in forming prior knowledge.
> > > Therefore, the current state-of-the-art (SOTA) methods for few-shot tabular learning utilize semi-few-shot learning approaches using unlabeled data samples or transfer tabular data to the text domain and employ Large Language Models.*
> > >
> > > [1] Borisov, V., Leemann, T., Seßler, K., Haug, J., Pawelczyk, M., & Kasneci, G. (2022). Deep neural networks and tabular data: A survey. IEEE Transactions on Neural Networks and Learning Systems.
> > >
> > > [2] Ryan, M. (2020). Deep learning with structured data. Simon and Schuster.
> > >
> > > **Q2, (minor) Since I have plenty of experience in visual few-shot learning, I deeply doubt the experiment with CLIP. As evidenced by Luo et. al [1], the visual encoder of CLIP performs extremely well on each of the diverse 10 datasets (including several natural image datasets), thus there is no reason that CLIP cannot perform better than a miniImageNet-trained model on table-based images. Besides, I do not see any value in replacing the backbones with ViT. What I mean is to use better pretrained models trained on massive datasets to see the relationship between the quality of the visual encoder and the few-shot tabular performance. Note that this is not a major problem to reject the paper, but doing this can better enhance the value of the paper.**
> > >
> > > - We believe there may have been a mistake in the experimental setup of our experiment using CLIP. We agree with your opinion that using a powerful visual encoder, including CLIP, can positively influence the performance of TablEye. However, we lack the time to conduct further experiments from multiple perspectives at this moment. We will proceed with additional experiments in the future and report the results.

---

> > > > ### Comment · Reviewer_6PPe · 2023-11-22
> > > >
> > > > Thanks for the response. I am satisfied with the additional related work which appropriately contextualizes the paper. I also understand the limited time for conducting more experiments which can be left to the near future. I thus increased the score to 6.
> > > >
> > > > Minor: I noticed that there is always no space between citations and texts, please check carefully and modify them.

---

> > > > > ### Comment · Reviewer_6PPe · 2023-11-23
> > > > >
> > > > > After another careful read of other reviewers' and Caspian's comments, I find that the proposed method lacks novelty and scalability, thus I decide to maintain the original score (5).

---

### Official Review · Reviewer_NMSd · 2023-10-30

**Soundness:** 2 fair
**Presentation:** 2 fair
**Contribution:** 2 fair
**Rating:** 5
**Confidence:** 4

**Summary:**

The authors present a new few-shot learning method for tabular data. By transforming the tabular data into image representations (_tabular images_), they hope to transfer prior knowledge that is readily available in the image domain onto the tabular task to improve results and make up for the scarcity of otherwise shared/prior information in the tabular domain. The argument is that in this way, proven methods from the well-explored image-based few-shot learning area can be leveraged to advance the area of tabular few-shot learning. The authors test their approach using two popular few-shot methods and four vision backbones on a variety of datasets.

**Strengths:**

### Originality & Significance:
The paper explores an interesting underlying idea to leverage information from a well-explored area (in this case the image domain) and transfer both prior knowledge and existing/proven algorithmic methods;

### Quality:
- Data: Authors experiment on different datasets and consider different important aspects: 1) feature diversity (categorical vs. numerical), 2) task diversity (n-way classification), 3) applicability to/relevance for ‘real-world’ applications, in this case medical data;
 - Architectures: experimentation with 4 different versions to gauge parameter and architectural influences;

I do however see severe weaknesses in most other parts, see the following.

**Weaknesses:**

While I do like the general underlying idea, there are several severe weaknesses present in this work – leading me to lean towards rejection of the manuscript in its current form. The two main areas of concern are briefly listed here, with details explained in the ‘Questions’ part:

### 1) Lacking quality of the “Domain Transformation” part
This is arguably the KEY part of the paper, and needs significant improvement in two points: Underlying intuition/motivation/justification,   as well as technical correctness and clarity.  There are several fundamental points that are unclear to me and require significant improvement and clarification; This applies to both clarity in terms of writing but, more importantly, to the quality of the approach and justifications/underlying motivations.
Please see the “Questions” part for details.

### 2) Lacking detail in experiment description:
Description of experimental details would significantly benefit from increased clarity to allow the user to better judge the results, which is very difficult in the manuscript’s current state; See "Questions" for further details.

**Questions:**

### Main questions regarding Domain Transformation part:

Technical parts:

-	Creating the (N,N) feature matrix R via Euclidean distance between N features -> What is the intuition behind this? Euclidean distance is symmetric (as squared), so isn’t the (N,N) matrix symmetric (if unranked) or has double-entries (if sorted/ranked)?
-	The authors then go on to state: “We also measure the distance and rank between N elements [..] to generate an (N,N) pixel matrix, denoted as Q.” -> What exactly is being compared/contrasted here? What ‘pixels’ are used here?
-	This is followed by another Euclidean distance between R and Q – Again, I am missing the intuition/justification behind this.
-	The authors claim that this then results in “a 2-dimensional image of size (Nr x Nc)”. How exactly is this obtained from computing the Euclidean distance between two (N,N) matrices?
---
Further details & justification:

-	How are the ranked features arranged to form a 2D ‘image’? This should significantly affect the way how ConvNets perform on them! More detail is required here.
-	Why would a ranking of the distances between features and pixels followed by rearrangement in any way resemble information presented in natural images? In natural images, the local relationship between pixels is defined by the occurrence of objects at a spatial location within the image. Why should a network pretrained on such data (in this case miniImageNet) be ‘useful’ to work on the artificially created tabular images? How do you overcome the (potentially significant) domain gap here? Or at least, what is the intuition behind it? (While the authors provide some insight in Figure 4, a 2D circle in t-SNE is not necessarily representative due to the hyperparameters involved in the projections); I'd invite the authors to further comment on this and their underlying intuitions.
---
-	Additionally: Since common CNNs take in RGB images (3 channels) but the authors create only images w/ 1 channel, they simply repeat the same image 3x for each channel – this seems like unnecessary overhead and simply engineered to fit existing input layers. If the created images are simply grayscale (as they seem to be according to Figure 1), wouldn’t it be more reasonable to pretrain the backbone on grayscale images?
-	In the introduction, the authors state that “features within tabular data have independent distributions and ranges, and missing values may be present.” Neglecting the missing values, how are the authors treating this challenge of different ranges? The Euclidean distance between features can largely vary if ranges differ, so how exactly are these values converted into image pixels which usually are defined within a fixed range of [0, 255] per channel?
---

### Experiments & Interpretation:

Table 1 aims to demonstrate the benefit of “Prior Knowledge Learnt from the image domain” -> I’d like the authors to further clarify the exact experimental setting that has been performed here:
- Are the experiments without image-pretraining simply trained on the tabular images?
- Or are they using a ‘randomly initialized’ backbone?
- Are the image-pretrained methods further fine-tuned on some tabular image data?

All this information will help the reader to better judge to which extend information is potentially ‘transferred’, what might be the risk of overfitting, etc.;

---

> ### Author Response · Authors · 2023-11-19
> **Rebuttal**
>
> **Technical parts:**
>
> **Q1. Creating the (N,N) feature matrix R via Euclidean distance between N features -> What is the intuition behind this? Euclidean distance is symmetric (as squared), so isn’t the (N,N) matrix symmetric (if unranked) or has double-entries (if sorted/ranked)?**
>
> - First, we have incorporated comments by adding detailed explanations to Section 3.2, "Domain Transformation," on page 4 and to "Algorithm 1" on page 14 of the paper. As described in the paper, we aimed to create an (n, n) feature matrix to intuitively represent the similarities among features of tabular data, similar to an adjacency matrix. Therefore, as you have understood, the (n, n) matrix is a symmetric matrix.
>
> **Q2. The authors then go on to state: “We also measure the distance and rank between N elements [..] to generate an (N,N) pixel matrix, denoted as Q.” -> What exactly is being compared/contrasted here? What ‘pixels’ are used here?**
>
> - Thank you for providing a detailed question. I recognize that the wording I used in that section did not clearly convey my intent, and we have revised section 3.2 on page 4 of the paper accordingly. As explained in the revised paper, when tabular data with n features is directly converted into a 2-dimensional image form, each pixel represents one of the n features. Therefore, the feature matrix calculates the distances between the coordinates of these n pixels. For instance, if there is tabular data where n=6=2*3, converting this data into a 2-dimensional image form would assign the 6 features coordinates like (0,0), (0,1), (0,2), (1,0), (1,1), (1,2). The feature matrix then calculates the distances between these coordinates, as illustrated in the figure.
>
> |   |   |   |   |   |   |
> |---|---|---|---|---|---|
> | 0 | 1 | 2 | 1 | √2 | √5 |
> | 1 | 0 | 1 | √2 | 1  | √2 |
> | 2 | 1 | 0 | √5 | √2 | 1  |
> | 1 | √2 | √5 | 0 | 1  | 2  |
> | √2 | 1 | √2 | 1 | 0  | 1  |
> | √5 | √2 | 1 | 2 | 1  | 0  |
>
> **Q3. This is followed by another Euclidean distance between R and Q – Again, I am missing the intuition/justification behind this.**
>
> - The feature matrix R represents the similarity between features when tabular data is simply reshaped into two dimensions. Q, on the other hand, signifies the distances between coordinates in an image of the same shape. Therefore, Q is a matrix of an image with the natural characteristic (spatial relation) where similar values are located near each other. Hence, we hypothesized that by calculating and reducing the distance between R and Q, we could impart spatial relations onto R. Indeed, our experiments confirmed that altering the order of features in R to minimize its distance from Q contributed to performance improvements in few-shot learning. The experimental results show that the method of directly converting tabular data into tabular images, termed 'direct', does not involve the process of reducing the distance between R and Q. It becomes evident that adding spatial relations indeed aids in enhancing performance.
>
> - If you look at the additional experiments added to Appendix G on page 16 of the paper, you will find a comparison of the performance of three methods: a direct method that manipulates shapes in 2D without considering spatial relations, a Graph method that represents the shapes as graphs, and our domain transformation method. The experimental results demonstrate that the presence or absence of spatial relations significantly impacts the performance of few-shot learning.
>
> **Q4. The authors claim that this then results in “a 2-dimensional image of size (Nr x Nc)”. How exactly is this obtained from computing the Euclidean distance between two (N,N) matrices?**
>
> - To provide a simple example, let's assume feature matrix R initially represents the similarity between (f1, f2, f3, f4) and (f1, f2, f3, f4), where f1, f2, f3, f4 are arbitrary features. If Nr=2 and Nc=2, the image formed would be [[f1, f2], [f3, f4]]. However, if the order of the features is changed, it can be seen as altering the similarity to that between (f3, f1, f4, f2) and (f3, f1, f4, f2). In this case, the image would become [[f3, f1], [f4, f2]]. This process is detailed in Algorithm 1 on page 14 of the paper.

---

> > ### Author Response · Authors · 2023-11-19
> > **Rebuttal(Cont)**
> >
> > **Further details & justification:**
> >
> > **Q5. How are the ranked features arranged to form a 2D ‘image’? This should significantly affect the way how ConvNets perform on them! More detail is required here.**
> >
> > - The method of forming a 2D image using ranked features is explained in conjunction with Q4. Additionally, more detailed information can be found in section 3.2 on page 4 of the paper and in Algorithm 1 on page 14.
> >
> > **Q6. Why would a ranking of the distances between features and pixels followed by rearrangement in any way resemble information presented in natural images? In natural images, the local relationship between pixels is defined by the occurrence of objects at a spatial location within the image. Why should a network pretrained on such data (in this case miniImageNet) be ‘useful’ to work on the artificially created tabular images? How do you overcome the (potentially significant) domain gap here? Or at least, what is the intuition behind it? (While the authors provide some insight in Figure 4, a 2D circle in t-SNE is not necessarily representative due to the hyperparameters involved in the projections); I'd invite the authors to further comment on this and their underlying intuitions.**
> >
> > - As you mentioned, pixels in natural images that are locally related tend to form objects of varying sizes and often share similar values. We believed that our domain transformation method could bridge this gap. For example, if you flatten a natural image to create n features and calculate the similarity among these n features, you can form an (n, n) matrix M. (M_ij represents the similarity between the i-th and j-th features, and M is symmetric.) Visualizing this normalized matrix M would likely show that areas in close proximity have high similarity, resulting in a natural appearance. This is similar to the pixel matrix Q generated from coordinates in domain transformation.
> >
> > - Therefore, by altering R to closely resemble Q through domain transformation, we were able to transform tabular data into a format that shares some characteristics with natural images. While still distinguishable to the human eye, from the perspective of a CNN, tabular images and natural images exhibit similar patterns. This finding underscores the adaptability of CNNs in recognizing patterns across different data modalities, and the potential of domain transformation in bridging the gap between disparate data domains.
> >
> > **Q7. Additionally: Since common CNNs take in RGB images (3 channels) but the authors create only images w/ 1 channel, they simply repeat the same image 3x for each channel – this seems like unnecessary overhead and simply engineered to fit existing input layers. If the created images are simply grayscale (as they seem to be according to Figure 1), wouldn’t it be more reasonable to pretrain the backbone on grayscale images?**
> >
> > - Thank you for the insightful question. Indeed, stacking 1 channel to create a 3-channel image does introduce computational overhead for image processing. However, the reason we experimented with 3-channel images is that we wanted to leverage as much information as possible in the process of learning prior knowledge from large image datasets. The table below shows the experimental results of learning prior knowledge from 1-channel grayscale images and performing prediction tasks on 1-channel tabular images. '1-ch' refers to 1-channel, and '3-ch' refers to 3-channel. The results indicate that 3-channel generally performed better in most cases. While there are a few instances where 1-channel closely matches or slightly outperforms 3-channel, there is a significant variation in performance in general scenarios. The reason for the superior performance of 3-channel images is not yet clear. We plan to explore and share more detailed reasons in future research. Additional experiments have been included on pages 14 and 15 of the paper.
> >
> > **Q8. In the introduction, the authors state that “features within tabular data have independent distributions and ranges, and missing values may be present.” Neglecting the missing values, how are the authors treating this challenge of different ranges? The Euclidean distance between features can largely vary if ranges differ, so how exactly are these values converted into image pixels which usually are defined within a fixed range of [0, 255] per channel?**
> >
> > - We performed min-max normalization on all tabular data to scale the values between 0 and 1. Then, we applied the domain transformation algorithm and finally multiplied all values by 255 to achieve an integer value in range between 0 and 255.

---

> > > ### Author Response · Authors · 2023-11-19
> > > **Rebuttal(Cont)**
> > >
> > > **Experiments & Interpretation:*
> > > **Q9. Table 1 aims to demonstrate the benefit of “Prior Knowledge Learnt from the image domain” -> I’d like the authors to further clarify the exact experimental setting that has been performed here:**
> > >
> > > *Are the experiments without image-pretraining simply trained on the tabular images?*
> > >
> > > *Or are they using a ‘randomly initialized’ backbone?*
> > >
> > > *Are the image-pretrained methods further fine-tuned on some tabular image data?*
> > >
> > > - Thank you for pointing out the unclear part of the experimental setup. In the experiment of Table 1, the "no img" condition refers to using a randomly initialized backbone to train only on the tabular images provided in the support set. Additionally, fine-tuning methods were not applied in any of the experiments. This information has been added to the description of Table 1 on page 7 of the paper.
> > >
> > >
> > > - *Before*: ‘No Img’ represents the condition where no prior knowledge learning has occurred in the image domain, whereas ‘Img’ denotes cases where prior knowledge has been acquired using mini-ImageNet.
> > >
> > >
> > > - *After*: ‘No Img’ represents the condition where no prior knowledge learning has occurred in the image domain. Randomly initialized backbone is applied and trained on a tabular image. ‘Img’ denotes cases where prior knowledge has been acquired using mini-ImageNet

---

> > > > ### Comment · Reviewer_NMSd · 2023-11-23
> > > > **Thank you for the provided answers**
> > > >
> > > > I'd like to genuinely thank the authors for providing detailed answers and making a number of improvements to the manuscript, I do appreciate the effort!
> > > > Having read the new manuscript version, my concerns about the missing details on the crucial part of domain transformation have been mostly resolved.
> > > > However, I feel that your paper still somewhat misrepresents your contribution to the domain transformation, the main parts of which are mostly introduced in Zhu et al.'s work as the direct quote from their manuscript shows:
> > > > > *To meet this challenge, we develop a novel algorithm, image generator for tabular data (IGTD), to transform tabular data into images by assigning features to pixel positions so that similar features are close to each other in the image. The algorithm searches for an optimized assignment by minimizing the difference between the ranking of distances between features and the ranking of distances between their assigned pixels in the image.*
> > > >
> > > > I further share the concerns of reviewer hBJa regarding the effectiveness of the approach.
> > > >
> > > > $\rightarrow$ Given these new insights, I have slightly raised my score to  *5: marginally below the acceptance threshold* but remain still critical.

---

> ### Author Response · Authors · 2023-11-22
> **Gentle Reminder**
>
> Dear Reviewers,
>
> I hope this message finds you well. As the deadline for the peer review of my academic paper is approaching tomorrow, I wanted to send a gentle reminder.
>
> Thank you very much for your time and dedication to this process. Should you have any additional comments or questions regarding the paper, please feel free to ask, and I will do my utmost to provide comprehensive and helpful responses.
>
> Warm regards,
>
> Authors

---

### Official Review · Reviewer_hBJa · 2023-10-31

**Soundness:** 3 good
**Presentation:** 2 fair
**Contribution:** 2 fair
**Rating:** 5
**Confidence:** 3

**Summary:**

In this paper, the authors delve into the domain of few-shot tabular representation learning by introducing a novel perspective—treating tabular data as images. They introduce a method called TablEye, which begins by converting tabular data into the image domain and subsequently harnesses image-based representations to enhance performance in few-shot tabular learning tasks. Notably, the experimental results showcase TablEye's efficacy as it outperforms existing methods such as TabLLM and STUNT in these tasks.

**Strengths:**

One of the notable strengths of this paper is the novel idea of utilizing image domain priors for few-shot tabular learning. This approach capitalizes on the inherent structure and relationships within image data to address the challenges of tabular learning, demonstrating its effectiveness in transferring knowledge to few-shot scenarios.

**Weaknesses:**

While TablEye represents a promising approach, it is not without its limitations. One concern is the potential scalability issues that might arise when dealing with tabular data possessing a substantial number of features. The transformation of tabular data into an image format could lead to image dimensions that are impractically large, which may hinder the method's scalability and efficiency. Additionally, the authors acknowledge that for heterogeneous tabular data, establishing meaningful spatial relationships within the transformed images can be a daunting task. This limitation suggests that the proposed method may not be a universally applicable solution for all tabular learning problems, especially those with highly diverse data structures.

**Questions:**

The paper raises intriguing questions regarding the choice of feature extraction techniques. While the primary focus of the paper lies in feature extraction using Convolutional Neural Networks (CNNs), the authors mention the possibility of utilizing pre-trained Vision Transformers (ViT). It prompts further exploration of whether ViT could serve as a viable alternative to CNNs for this specific application. The underlying assumption that inductive bias plays a crucial role in the success of TablEye raises the question of whether ViT, with its distinct characteristics, would be as effective in leveraging this bias.

Furthermore, the paper highlights the potential challenge of handling tabular data with an exceedingly large number of features. It is worth considering how a conventional CNN architecture, or even alternative methods, could adapt to accommodate such datasets while maintaining computational efficiency. This consideration adds an interesting dimension to the discussion about the method's scalability and practicality in real-world applications.

---

> ### Author Response · Authors · 2023-11-19
> **Rebuttal**
>
> **Q1. The paper raises intriguing questions regarding the choice of feature extraction techniques. While the primary focus of the paper lies in feature extraction using Convolutional Neural Networks (CNNs), the authors mention the possibility of utilizing pre-trained Vision Transformers (ViT). It prompts further exploration of whether ViT could serve as a viable alternative to CNNs for this specific application. The underlying assumption that inductive bias plays a crucial role in the success of TablEye raises the question of whether ViT, with its distinct characteristics, would be as effective in leveraging this bias.**
>
> - We conducted experiments using the CLIP pre-trained ViT encoder and also using ViT as a backbone with the Proto-layer as a classifier. Initially, we directly used CLIP's pre-trained ViT encoder as a backbone for few-shot classification. However, CLIP demonstrated almost no learning in all cases. We believe this is because pre-trained models are already trained in a specific direction in the case of CLIP, to narrow the distance with similar texts. When we used ViT as the backbone and trained it with prior knowledge from the mini-ImageNet dataset, T-P-V showed quite good performance. Although it was lower than other TablEye methods, this suggests that the characteristics of ViT can utilize the inductive bias of TablEye. This finding highlights the potential adaptability of ViT in different learning contexts, especially in tasks like few-shot learning where leveraging prior knowledge and inductive biases is crucial. We have added these experimental results to Appendix H on page 16 of the paper.
>
> **Q2. Furthermore, the paper highlights the potential challenge of handling tabular data with an exceedingly large number of features. It is worth considering how a conventional CNN architecture, or even alternative methods, could adapt to accommodate such datasets while maintaining computational efficiency. This consideration adds an interesting dimension to the discussion about the method's scalability and practicality in real-world applications.**
>
> - Thank you for raising the crucial issue of scale-up. We considered whether TablEye could function in scenarios with a large number of features. For TablEye to work, domain transformation must occur, and the backbone and classifier must be trainable. Since image models can adapt to the size of the input image, TablEye can operate as long as domain transformation is feasible. Given that the time complexity of Domain Transformation is O(n^2), this process will likely be the bottleneck when dealing with a large number of features. However, given enough time, it is still operational. We agree that this is an important issue worth further exploration. In future work, we plan to investigate new algorithms that can maintain the performance of domain transformation while reducing its time complexity. This would significantly enhance the scalability of TablEye, especially in large-scale applications.

---

> ### Author Response · Authors · 2023-11-22
> **Gentle Reminder**
>
> Dear Reviewers,
>
> I hope this message finds you well. As the deadline for the peer review of my academic paper is approaching tomorrow, I wanted to send a gentle reminder.
>
> Thank you very much for your time and dedication to this process. Should you have any additional comments or questions regarding the paper, please feel free to ask, and I will do my utmost to provide comprehensive and helpful responses.
>
> Warm regards,
>
> Authors

---

> > ### Comment · Reviewer_hBJa · 2023-11-22
> > **Response**
> >
> > Thanks for the authors for the rebuttal.
> >
> > After reading the rebuttal, I am still questionable about the application of TablEye. For example, I believe TablEye rely on the representational knowledge of pre-trained Image encoder, but using better representations (e.g., CLIP encoder) do not work better than CNN trained on mini-ImageNet, which questions the effectiveness of proposed approach. Also, I think the proposed method would lose efficiency as the number of features increases. Lastly, from the public comment by Capsian, there is a prior work that used the idea of using image feature for tabular learning, which weakens the novelty of proposed method. Therefore, I keep my score.

---

### Official Review · Reviewer_TNZH · 2023-11-02

**Soundness:** 3 good
**Presentation:** 3 good
**Contribution:** 3 good
**Rating:** 8
**Confidence:** 4

**Summary:**

The paper presents TablEye, a novel framework for few-shot tabular learning. To overcome the limit of forming prior knowledge for tabular data, TablEye utilizes a two-stage process, transforming tabular data into tabular images and learning prior knowledge from labeled image data. The paper reports improved performance and applicability to medical datasets. Overall, I vote for accepting. TablEye introduces a novel approach to few-shot tabular learning, which is a relatively underexplored area in research. There are only several techniques for this task and they still have some constraints. The paper offers innovative solutions, using prior image knowledge through a few-shot learning method, and demonstrates clear performance improvements.

**Strengths:**

TablEye addresses a challenging problem in few-shot tabular learning using a unique approach. Few-shot tabular learning is a relatively new and underserved area, and the paper contributes to this domain. TablEye introduces an innovative approach to few-shot tabular learning by bridging the gap between tabular and image data domains.

The paper provides evidence of TablEye’s effectiveness through a series of experiments. TablEye consistently outperforms existing methods in multiple scenarios, including 1-shot and 4-shot learning. The use of a significantly smaller model size compared to alternatives and less constraints on datasets are also strong points.

The paper demonstrates the applicability of TablEye to real medical datasets, which implies its potential value in practical applications, especially in domains where accurate few-shot tabular learning is crucial.

**Weaknesses:**

While the paper presents positive results, it lacks detailed discussions regarding the differences in dataset performances. A more in-depth analysis of why certain structures perform better on specific datasets would provide a deeper understanding of TablEye's strengths. It seems that the improvement in performance is partly based on the variety of structures since none of them perform well in most of the experiments.

The experiments of T-M-C2, T-M-C3, T-M-C4 on comparison with TabLLM and in the context of medical results are lacking.
Lack of Detailed Implementation: The paper offers an overview of the framework but lacks detailed implementation specifics. It would be better if you could present the structure of classifiers and also some equations or pseudo-code for all the parts of the model, especially the domain transformation.

Additional Context for Few-Shot Learning: Providing a brief introduction to the general few-shot learning problem, its significance, and existing challenges would be beneficial for readers unfamiliar with the field.

**Questions:**

Could you please provide more details on the differences in performances across different datasets, particularly explaining why some structures perform better on specific datasets? This would help in understanding TablEye's strengths and limitations better. Could you explain why STUNT performs better in the dataset Karhunen? Similarly, why in some cases Conv2 is better than Conv4, for example, the datasets Optdigits and Karhunen?

Moreover, Could you please provide experiments on T-M-C2, T-M-C3, T-M-C4 on comparison with TabLLM and in the context of medical results? Are there any recommendations or guidelines for selecting the most appropriate structure when using datasets?

Could you offer more detailed implementation specifics? The figures, equations, or pseudo-code can help understand each part of the model better, especially the domain transformation, which was kind of hard to understand at first.
For the part about repeating the matrix, have you tried other methods like resizing or padding besides tilling when dealing with the matrix? Is tilling the best solution?

And at last, it would be better if you could present briefly the structure of the classifiers you used.

---

> ### Author Response · Authors · 2023-11-19
> **Rebuttal**
>
> **Q1. Could you please provide more details on the differences in performances across different datasets, particularly explaining why some structures perform better on specific datasets? This would help in understanding TablEye's strengths and limitations better. Could you explain why STUNT performs better in the dataset Karhunen? Similarly, why in some cases Conv2 is better than Conv4, for example, the datasets Optdigits and Karhunen?**
>
> - Generally, TablEye demonstrates consistent high performance across numeric, categorical, and mixed values. TabLLM excels particularly with categorical values, while STUNT tends to perform exceptionally well with numeric values. In the case of STUNT, being a semi-supervised learning approach, the performance improvement was significantly noticeable over the baseline (XGB) when there was an abundance of unlabeled data and a higher number of features (collected information for prediction).
>
> - A trend was observed where performance tended to reverse with an increasing number of features. This was not only evident in datasets like optdigits (64 features) and karhunen (64 features) but also in Cancer (32 features), Heart (12 features), and Lung (16 features). The authors speculate that this might be due to the training epochs being uniformly limited to 100. In simpler structures (small images), convergence up to conv4 leads to superior performance, whereas in more complex structures (large images), Conv3 or Conv2 tends to perform better. Taking into consideration the reviewer's comments, the authors agree on the necessity to experiment by increasing the number of training epochs and comparing accuracy, and plan to conduct and report this in the future.
>
> **Q2. Moreover, Could you please provide experiments on T-M-C2, T-M-C3, T-M-C4 on comparison with TabLLM and in the context of medical results?**
>
> - We conducted experiments on T-M-C2, T-M-C3, and T-M-C4 with TabLLM and in the context of medical results These findings are reflected in Table 2 on page 7 and Table 7 on page 15 of the paper. The results showed that methods using the MAML-layer achieved higher numbers in some cases compared to previous figures.
>
>
> **Q3. Are there any recommendations or guidelines for selecting the most appropriate structure when using datasets?**
>
> The type of recommendation or guidance needed varies depending on the data and learning environment. If there are a few features, a complex structure (like conv4) might converge with a small amount of training, showing a high possibility of achieving high accuracy. However, if there are many features and training is minimal, a complex structure may not learn effectively, and a simpler structure could be more beneficial. If there are many features but sufficient resources for adequate training, we recommend without hesitation using a complex structure.
>
> | Epoch Count / Number of Features | Few      | Many                                      |
> |----------------------------------|----------|-------------------------------------------|
> | Few                              | Complex  | Simple                                    |
> | Many                             | Complex  | Unknown, but complex structure recommended |

---

> > ### Author Response · Authors · 2023-11-19
> > **Rebuttal (Cont)**
> >
> > **Q4. Could you offer more detailed implementation specifics? The figures, equations, or pseudo-code can help understand each part of the model better, especially the domain transformation, which was kind of hard to understand at first. For the part about repeating the matrix, have you tried other methods like resizing or padding besides tilling when dealing with the matrix? Is tilling the best solution?**
> >
> > - For providing detailed implementation insights, we have published pseudocode covering domain transformation. This content has been added to the Appendix D Algorithm section on pages 14 and 15 of the paper. We also enhanced the section 3.2 on domain transformation to facilitate a deeper understanding of the details among readers. Regarding the repetition of elements in the matrix, what we actually repeat are the elements contained within the matrix. Visually, this has an effect akin to magnifying a small image, which I believe is what you referred to as resizing. We initially experimented with filling a larger matrix with multiple smaller matrices. However, this approach resulted in very small edges and ambiguous boundaries, making it challenging for CNNs to extract meaningful features. Similarly, padding around the matrix didn't yield good results, especially when the base matrix size was very small, like (8,8). Among the intuitively conceivable methods, we believe the one we employed demonstrated the best performance.
> >
> > **Q5. And at last, it would be better if you could present briefly the structure of the classifiers you used.**
> >
> > - The Proto-layer algorithm computes prototypes (prototypes) for each class using support images and classifies query images based on these. The model is trained by calculating the cross-entropy loss between the classified results and the actual target. The MAML-layer algorithm is a process of fine-tuning the model to quickly adapt to a given task. This process is performed iteratively, and in each iteration, the model is updated based on the loss between the model's prediction and the actual target. We have added detailed descriptions of the classifier's structure in pseudo-code in Appendix D Algorithm on pages 13 and 14 of the paper.
> >
> > - Also, I agree with your opinion that some readers might not be familiar with few-shot learning in the process of understanding the classifier's algorithm. Therefore, we have added background on few-shot learning for readers who are not familiar with it in Appendix A on page 12 of the paper.

---

> ### Author Response · Authors · 2023-11-22
> **Gentle Reminder**
>
> Dear Reviewers,
>
> I hope this message finds you well. As the deadline for the peer review of my academic paper is approaching tomorrow, I wanted to send a gentle reminder.
>
> Thank you very much for your time and dedication to this process. Should you have any additional comments or questions regarding the paper, please feel free to ask, and I will do my utmost to provide comprehensive and helpful responses.
>
> Warm regards,
>
> Authors

---

### Public Comment · ~Caspian1 · 2023-11-11
**Discuss More  about one Related Work**

Thanks for the good work. You reference a paper [1] with limited discussion. The process of converting tabular data to images through aligning spatial relationships is very similar to this paper. The novelty of TABLEYE may not be fully reflected in this aspect. Could you please provide a more detailed comparison with this relevant work?

[1] Yitan Zhu, et al. Converting tabular data into images for deep learning with convolutional neural networks. Scientific reports, 11(1):11325, 2021.

---

> ### Author Response · Authors · 2023-11-21
> **Public Response**
>
> Thank you for your interest in our research. Our domain transformation was inspired by the spatial relations in IGTD. However, IGTD is entirely based on supervised learning and does not function properly in a few-shot setting with a limited number of samples. Therefore, we modified the calculation of the feature matrix in IGTD by using feature names to compute a more distinct feature distance. Also, IGTD only stores images as simple black and white. We have a mechanism that adds duplicate values and stores the images in a specific size (3,84,84). For more details, please refer to Section 3.2 on domain transformation on page 4 of the paper and Algorithm 1 in Appendix D on page 14.

---

### Author Response · Authors · 2023-11-21
**General Response**

All authors appreciate your thoughtful reviews and every word, even if it is a minor issue. All your comments help improve the quality of our paper. We have strived to conduct as many experiments as possible to fully incorporate your comments. Below is a summary of the detailed aspects of our paper that have been revised. The PDF file of the paper has been updated with the revised version, so please refer to the paper for more detailed information. If you have any additional question or advice to improve ours, please do not hesitate to leave your comments.

***Summary for Revision***

**In section 2. Related Work(page 2 and 3 of the paper)**

- We have added context that progresses from tabular learning to few-shot learning and then to few-shot tabular learning.

**In section 3.2 Domain Transformation(page 4 of the paper)**

- We have provided a more detailed explanation of the feature matrix and pixel matrix in the domain transformation process, and included detailed mathematical formulas.

**In section 3.3 Learning Prior Knowledge(page 5 of the paper)**

- At the end of the paragraph, we added the sentence, “Algorithm 2, 3 at Appendix D explain the detailed process of the backbone and classifier.” In Algorithm 2 and 3 of Appendix D, you can find a detailed explanation of the working principles of the classifier.

**In section 4.2 Ablation Study Table 1(page 7 of the paper)**

- In Table 1's experiments, we modified the title of the table to provide a detailed explanation of the settings 'no img' and 'img'.
    - *Before*: ‘No Img’ represents the condition where no prior knowledge learning has occurred in the image domain, whereas ‘Img’ denotes cases where prior knowledge has been acquired using mini-ImageNet.
    - *After*: ‘No Img’ represents the condition where no prior knowledge learning has occurred in the image domain. Randomly initialized backbone is applied and trained on a tabular image. ‘Img’ denotes cases where prior knowledge has been acquired using mini-ImageNe

**In section 4.3 COMPARISON RESULTS WITH TABLLM Table 2 (page 8 of the paper)**

- We have added the experimental results for T-M-C2, T-M-C3, and T-M-C4, which utilized the MAML classifier.

**In section 4.5 MEDICAL DATASETS FEW-SHOT RESULTS(page 15 of the paper)**

- This section has been moved to Appendix F due to space constraints. Additionally, we have added the experimental results for T-M-C2, T-M-C3, and T-M-C4, which utilized the MAML classifier.

**In section Appendix A. FEW-SHOT LEARNING BACKGROUND (page 12 of the paper)**

- We added this section to help users who are unfamiliar with few-shot learning gain a deeper understanding of the problem, including the classifier's algorithm. In this section, we briefly summarize the definition and issues of the few-shot learning problem, referencing several few-shot learning survey papers.

**In section Appendix D. ALGORITHM (page 14 of the paper)**

- Many reviewers asked questions about the detailed principles of domain transformation, especially in TablEye. To address these inquiries, we added this section to explain the detailed working processes of domain transformation and the classifier. In this section, we use pseudo code to provide a detailed explanation of both the domain transformation and the classifier.

**In section Appendix E. INPUT IMAGE CHANNEL SELECTION (page 13 of the paper)**

- There were questions about the overhead of using color images instead of black and white images as a result of domain transformation. To empirically demonstrate why we use color images instead of black and white images throughout the TablEye framework, we added this section. In this section, we detailedly compare the performance of training tabular images and mini-ImageNet in black and white images (1-channel) versus color images (3-channel). The experimental results showed that using color images (3-channel) generally yields higher performance than black and white images (1-channel).

**In section Appendix G. TABLE TO IMAGE CONVERSION METHOD (page 16 of the paper)**

- We added this section to provide a baseline for domain transformation. In this section, we compare the results of using TablEye with two different methods: directly transforming tabular data into 2D images without our proposed domain transformation method, and visualizing it in the form of graphs. The experimental results showed that the performance of TablEye is significantly higher when using our proposed domain transformation method.

**In section Appendix H. POTENTIAL OF VISION TRANSFORMER BACKBONE (page 16 of the paper)**

- We added this section to demonstrate the feasibility of using a vision transformer (ViT) as the backbone for TablEye. We measured the performance of TablEye when using ViT as the backbone and a Proto-layer as the classifier, and confirmed that ViT is indeed a viable option. This part is intended to be included in future, more developed research.

---

### Author Response · Authors · 2023-11-23
**Acknowledgment of Reviewer Feedback**

Dear Reviewers,

I would like to express my sincere gratitude for the valuable comments and insights you have provided on my research paper. Your expertise and thoughtful feedback are greatly appreciated.

I respect and acknowledge each point you have raised. The comments have been incredibly helpful and have offered new perspectives that I had not previously considered. I am committed to incorporating these insights into my future research endeavors.

Thank you once again for your constructive feedback and for the time and effort you have invested in reviewing my work.

Best regards,

Authors

---

### Meta-Review · Area_Chair_ExJi · 2023-12-05

**Metareview:**

This paper proposes an approach to few-shot tabular data classification by viewing tabular data as images.  This work has some exciting empirical results, and with some work, it could be a strong paper.  The reviewers raised several issues, notably regarding scalability and novelty of the contribution.  I also want to point out that this work lacks discussion of and comparison to strong tabular neural networks, such FT-Transformer, SAINT, and more recent ones.  It also lacks comparison to TabPFN, which can perform few-shot learning on tabular data and has achieved state-of-the-art performance on small datasets.  Moreover, the work only uses a relatively small number of datasets.  Therefore, I’m inclined to reject this paper, but I encourage the authors to beef up their promising work.

**Justification For Why Not Higher Score:**

The work is still missing a lot of important evaluations and baselines.  The current evaluations are essentially preliminary.

**Justification For Why Not Lower Score:**

N/A

---

### Decision · Program_Chairs · 2024-01-16

Reject